# Exacerbated fires in Mediterranean Europe due to anthropogenic warming projected with non-stationary climate-fire models

Marco Turco [1], Juan José Rosa-Cánovas [2], Joaquín Bedia [3,4], Sonia Jerez [2], Juan Pedro Montávez [2], Maria Carmen Llasat [1] & Antonello Provenzale[5]

The observed trend towards warmer and drier conditions in southern Europe is projected to continue in the next decades, possibly leading to increased risk of large fires. However, an assessment of climate change impacts on fires at and above the 1.5 °C Paris target is still missing. Here, we estimate future summer burned area in Mediterranean Europe under 1.5, 2, and 3 °C global warming scenarios, accounting for possible modifications of climate-fire relationships under changed climatic conditions owing to productivity alterations. We found that such modifications could be beneficial, roughly halving the fire-intensifying signals. In any case, the burned area is robustly projected to increase. The higher the warming level is, the larger is the increase of burned area, ranging from ~40% to ~100% across the scenarios. Our results indicate that significant benefits would be obtained if warming were limited to well below 2 °C.

[1] Department of Applied Physics, University of Barcelona, 08028 Barcelona, Spain. [2] Regional Atmospheric Modeling Group, University of Murcia, 30100 Murcia, Spain. [3] Predictia Intelligent Data Solutions, 39005 Santander, Spain. [4] Santander Meteorology Group, Department of Applied Mathematics and Computing Science, University of Cantabria, 39005 Santander, Spain. [5] Institute of Geosciences and Earth Resources (IGG), National Research Council (CNR), 56124 Pisa, Italy. Correspondence and requests for materials should be addressed to M.T. (email: turco.mrc@gmail.com)

The Paris Agreement of the United Nations Framework Convention on Climate Change (UNFCCC), signed in December 2015, aims "to hold the increase in the global average temperature to below 2 °C above preindustrial levels and to pursue efforts to limit the temperature increase to 1.5 °C…". Following the invitation from the UNFCCC, the Intergovernmental Panel on Climate Change (IPCC) is preparing a report on the impacts of global warming of 1.5 °C above pre-industrial levels (https://www.ipcc.ch/report/sr15/) to be published in 2018[1]. To provide updated information for the proposed IPCC special report, recent research efforts have significantly boosted our knowledge on the risks at 1.5 and 2 °C of warming, focusing on climate extremes[2] and the relevant impacts on agriculture[3], power generation[4], ecosystems[5], and hydrology[6]. However, the translation of ambitious warming targets into impacts on future wildfires remains to be studied.

Mediterranean Europe is a relevant region for such an analysis because fires frequently burn across this area, causing severe economic and environmental damage, including loss of lives, infrastructures, and ecosystem services such as carbon sequestration and the provisioning of raw materials, with an average of approximately 4500 km$^2$ burned every year[7–9]. For instance, the fire season in 2017 was severe in many regions of Southern Europe, with large wildfires in southern France, Italy, Portugal, and Spain associated with unusually intense droughts and heatwaves. These fires caused extensive economic and ecological losses and even human casualties[10].

Under changing climate conditions, several possible pathways of wildfire response can be identified depending on the magnitude of climate change, as well as on differences in how fires, vegetation, and humans respond to such changes[11]. Several studies support the hypothesis that in Southern Europe, summer drought conditions and high temperatures are primary drivers of the inter-annual variability of fires[9,12–21]. Previous works using a variety of approaches of increasing complexity, from correlation-based models[22–24] to process-based models[25,26], consistently indicate that fire risk is expected to increase in the future. To date, process-based models have been unable to reproduce the observed fire evolution[27,28] and show a large spread in future fire projections between models[26], while statistical analyses for the whole of Mediterranean Europe are still relatively scarce[9,18,22].

Previous empirical (data-driven) studies have derived models based on the sensitivity of fires to interannual climate variability, usually assuming the climate-fire relationship to be stationary over time. However, it is expected that a warmer and drier climate can affect wildfire activity not only by leading to more favourable conditions for burning but also by modifying the structure of the fuel (in terms of availability and continuity) to be burned. In other words, the nature of the fire climate links may change over time and relatively small variations in future climates could lead to drastic shifts in fire activity because of productivity alterations. Only if the direct effect of climate change in regulating fuel moisture (e.g., drier and warmer conditions increase fuel flammability leading to larger fires) continues to be dominant with respect to the indirect effect on fuel load and structure (e.g., drier and warmer conditions limit fuel availability), fire risks will increase[7,9,29–32] as the climate becomes warmer and drier[33,34].

In recent decades, changes in climate and other environmental and socioeconomic factors have significantly affected both fire regimes[35,36] and fire-climate links[37–42]. In particular, Pausas and Paula[42] show that in Mediterranean ecosystems, fuel determines the fire-climate relationship as wet and productive regions are more sensitive to flammable conditions than dry regions. The results obtained in these studies indicate the importance of considering the non-stationary nature of fire-climate relationships to obtain more realistic fire projections as climate change

may drive fuel structure changes and consequently modify the climate-fire relationship.

The aim of this paper is to explore the fire response in an ensemble of state-of-the-art regional climate projections (RCM) in Mediterranean Europe at 1.5, 2, and 3 °C of mean global warming. We built an ensemble of regional-scale models linking climate and summer burned area, and then we projected these relationships for different climate scenarios with and without taking into account how the long-term impact of climate on fuel productivity might affect the climate-fire relationships. Despite there are several sources of uncertainty, which are larger for longer time horizons, a consistent pattern emerges from the analysis of the available data, supporting the robustness of the results. Overall, we found that the projected increase in drought conditions leads to larger burned area values and that limiting global warming to 1.5 °C can strongly reduce the increase of burned area.

## Results

**Defining the climate-fire model.** A recent study[9] has shown that the area burned by summer fires is directly associated with same-summer drought conditions in most sub-regions of Mediterranean Europe. The approach discussed here builds upon this result and explores the relationship between drought indicators and fires through a statistical model. The influence of climate on BA in summer months (June, July, August and September; JJAS) is considered through the use of the standardised precipitation evapotranspiration index (SPEI[43]) as a climate indicator/predictor. For each sub-region of the domain (i), we express the possible link of year-to-year (t) changes in summer fires with the SPEI using the following model:

$$\log[\text{BA}(i,t)] = \beta_1(i) + \beta_2(i) \cdot SPEI_{sc,m}(i,t) + \beta_3(i) \atop \cdot T(t) + \varepsilon(i,t) \tag{1}$$

where BA($i,t$) is the BA in the ith eco-region and summer $t$; $\beta_1$ is the intercept; $\beta_2$ represents the sensitivity of BA in each region to dry conditions as summarised by the SPEI; $\beta_3$ is the coefficient of the time term $T$ (in years) that characterises the temporal trends of BA, thus taking into account the possible influence of slowly changing factors over the study period; and $\varepsilon$ is a stochastic noise term that captures all other (neglected) processes that influence BA other than SPEI and $T$. Drought conditions are measured by the SPEI indices aggregated in multi-month values, $SPEI_{sc,m}$, where $m$ is the month for which the SPEI is computed (which we allow to vary from previous spring to coincident summer months) and $sc$ is the time scale (number of months) used to compute the SPEI (we consider periods of 3, 6, and 12 months; for instance, $sc = 3$ corresponds to the precipitation, PRE, and potential evapotranspiration, PET, anomalies accumulated over the 3 months $m$-2, $m$-1, and $m$; see the methods section).

The SPEI-BA model shows good performance in reproducing the observed BA variations in most of the domain, with a spatially averaged correlation of 0.68 (and 0.54 obtained through a leave-one-out cross-validation). This result suggests that parsimonious regression models are able to explain at least some of the main processes determining the effects of climate variability on the Mediterranean summer BA, thus supporting their use in estimating fire response to different climate change projections (see Supplementary Figs. 1–3 for more details on the models parameters and skill and Supplementary Table 1 for an exact definition of the model for each region).

In Eq. 1, the term $T$ represents the linear temporal trends of the fire variable resulting from both anthropic effects (such as a gradual increase in fire management effort) and environmental/

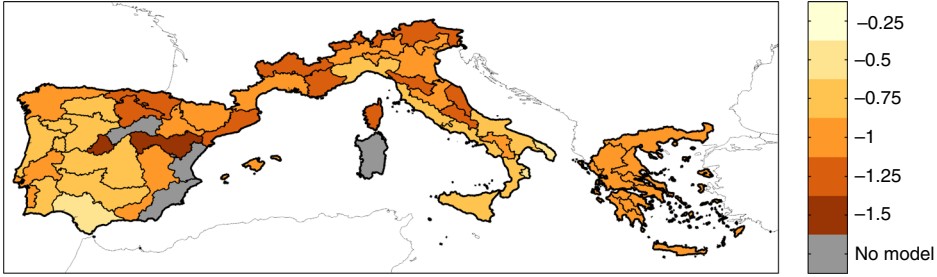

**Fig. 1** Sensitivity of burned area to SPEI variations. This sensitivity is represented by the coefficients $\beta_2$ of Eq. 1

| Table 1 Empirical climate-$\beta_2$ models (Eq. 2) considering different (temporally averaged) climate variables (first column) | | | |
|---|---|---|---|
| **Climate variable** | $\gamma_1$ | $\gamma_2$ | **Correlation** |
| $Ty$ | −1.62 | 0.057 | 0.64 |
| $Ts$ | −2.02 | 0.057 | 0.63 |
| $PREs$ | −0.68 | −0.00100 | 0.50 |
| $P\text{-}PETs$ | −1.05 | −0.00050 | 0.46 |
| $P\text{-}PETy$ | −0.90 | −0.00023 | 0.37 |
| $PREy$ | −0.61 | −0.00032 | 0.31 |

Note: Regression parameters are reported in the second and third columns, while the correlation between simulated and observed $\beta_2$ values is indicated in the last column

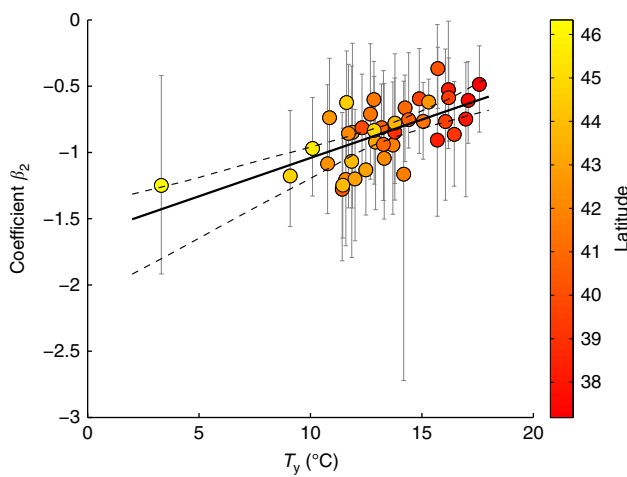

**Fig. 2** Relationship between the long-term average of annual temperature ($T_y$) versus the sensitivity of burned area to SPEI ($\beta_2$) for the different eco-regions. The colours of the points indicate the latitude of the centroid of the sub-region. Grey bars enclose 95% confidence intervals of the individual $\beta_2$ values. The black line indicates the best linear fit, while dashed lines indicate the 95% confidence interval of the linear regression model of Eq. 2

climatic changes. These trends are negative and significant in several regions (Supplementary Fig. 3; see also refs.[8,28] for more details). That is, in the past few decades, the measured trend of BA in Mediterranean Europe has generally been steady or negative, while drought conditions have increased[44,45]. These opposite trends suggest that management actions have so far counterbalanced the climatic trend[23,46].

The response of BA to SPEI variations (i.e., the parameter $\beta_2$, that is the fingerprint of climate on BA), is negative (Fig. 1). Because negative SPEI values correspond to hot and dry conditions, this result indicates that overall, the mechanism by which drought affects BA is straightforward: warmer and drier summers lead to larger fires. However, the drought-fire relationship is more complex. Already by visual inspection of Fig. 1, we observe that the generally higher absolute values of the parameter $\beta_2$ (i.e., a higher BA sensitivity to SPEI variation) are in the northern region. The statistical analysis that follows provides a confirmation. To find which climate variables better explain the spatial variation of $\beta_2$, we tested for several candidates, such as the long-term mean temperature (T), PRE, PET, and the water balance PRE-PET, considering the data aggregated both at annual and at summer scales in the following model:

$$\beta_2(i) = \gamma_1 + \gamma_2 \cdot X(i) + \varepsilon(i) \qquad (2)$$

where $X(i)$ is the temporally averaged value of the chosen climate variables in the eco-region ($i$), $\gamma_1$ is the intercept, $\gamma_2$ is the coefficient of the climate term $X$, and $\varepsilon$ is a spatially uncorrelated stochastic noise term.

Several potential models (with T, PRE, or PRE-PET at the annual or summer scale) show reasonable skill in reproducing $\beta_2$. Table 1 reports the results for which the correlation between simulated and observed values is statistically significant and the null hypothesis of negligible spatial autocorrelation of the residuals is accepted (using Moran's $I$ test). The best performing model is based on the long-term annual mean temperature ($T_y$); thus, we consider this variable in the following analysis. However,

because the choice between the different models may be critical, we also tested the sensitivity of the outcomes to model selection and estimated BA using all the models in Table 1.

The relationship between the long-term average of annual temperature ($T_y$) versus the sensitivity of BA to SPEI ($\beta_2$) for the different eco-regions suggests that in (northern) colder, wetter and more productive regions (where $T_y$ shows lower values), drought plays a more prominent role for BA than in (southern) drier regions (where $T_y$ shows larger values; Fig. 2). This result is in line with the results obtained for vegetation-fire-climate relationships in Mediterranean areas[42], which shows that the sensitivity of fire activity to dry periods is larger in productive zones.

We interpret this spatial variation as a surrogate for potential non-stationarity in the BA-SPEI links. That is, the value of $\beta_2$ in southern regions may serve as an analogue for the BA-SPEI relation in the northern regions that will experience an increase in temperature. In other words, if the value of $T_y$ in northern regions increases up to the values observed in southern areas, the current changes in these latter regions can provide a hint on the future evolution in northern areas. Note that this similar adjustment/ adaptation strategy is widely used to analyse the effects of climate change on the economy and agriculture (see, e.g.,[47–49]), but to the best of our knowledge, this approach has never been applied to study the impact on fires as done here.

**Fire projections.** To explore the behaviour of future BA, we proceed as follows. First, we consider the stationary SPEI-BA

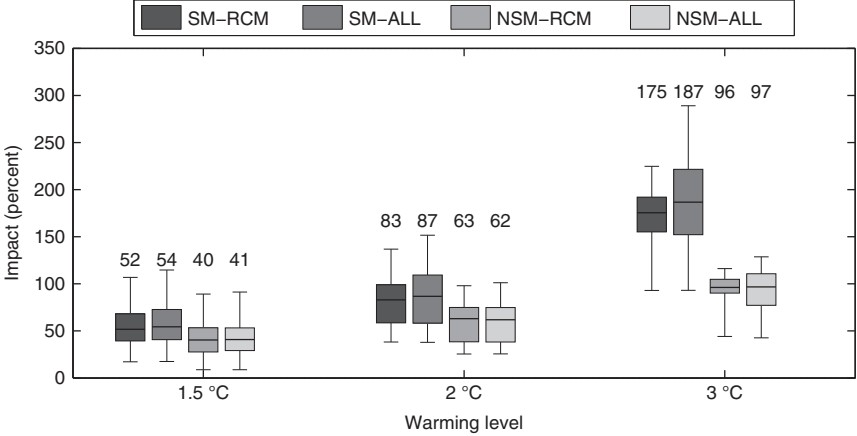

**Fig. 3** Burned area changes in Mediterranean Europe with the stationary and the non-stationary models. Burned area changes (in %) are shown for each warming level considering the stationary model SM (i.e., using Eq. 3) and the non-stationary model NSM (i.e., using Eq. 4). Boxplots show the uncertainty from the ensemble of RCM projections (SM-RCM or NSM-RCM) and accounting for both RCM and regression model uncertainties (1000 bootstrap replications × the ensemble of RCMs; SM-ALL or NSM-ALL). The median is shown as a solid line, the box indicates the 25–75 percentile range, while the whiskers show the 2.5–97.5 percentile range. The numbers above the boxes indicate the median values

model (hereinafter SM):

$$\log[\mathrm{BA}(i,t)]_{\mathrm{climate}} = \beta_1(i) + \beta_2(i) \cdot SPEI_{sc,m}(i,t) + \varepsilon(i,t) \quad (3)$$

Then, to take into account the potential changes in the SPEI-BA links, we redefine $\beta_2$ in Eq. 3 to follow the $T_y$-$\beta_2$ relationship provided by Eq. 2 and Table 1, in what we call the non-stationary model (hereinafter, NSM):

$$\log[\mathrm{BA}(i,t)]_{\mathrm{climate}} = \beta_1(i) + (\gamma_1 + \gamma_2 \cdot X(i)) \\ \cdot SPEI_{sc,m}(i,t) + \varepsilon(i,t) \quad (4)$$

The models in Eqs. 3 and 4 explicitly depend only on climatic variables. Both models may be useful, even when the assumption of stationarity in the SM model does not hold true. Indeed, comparing SM with NSM projections provides a measure of the contribution of the climate-fire link changes to the projected BA.

At this point, we drive the models in Eqs. 3 and 4 with RCM projections, selecting the temporal windows where the global mean temperature increase is 1.5, 2, or 3 °C. The spatially averaged BA changes for different warming levels and for different model specifications are displayed in Fig. 3. Four main conclusions can be drawn from this analysis.

First, a robust increase in BA is projected over Mediterranean Europe. Second, this increase is much higher for 2 °C (with values between 62 to 87% depending on the model specifications) and 3 °C of global warming (with values between 96 to 187%) compared to the 1.5 °C target (with values between 40 to 54%). Third, the results indicate that NSM (non-stationary) models generally led to lower impacts, especially for larger temperature variations. For the +3 °C case, BA shows increases of 175 to 187% (depending on considering only the RCMs or the model + RCMs spread) with SM and of +96% to +97% with the NSM approach. Finally, we note that the overall uncertainty is dominated by the RCM spread rather than by the uncertainties related to the climate-fire model. Indeed, there are only minor differences between the (N)SM-RCM boxes (RCM model uncertainty, as given by the multimodel spread) and between the (N)SM-ALL boxes (RCM + model parameter estimation uncertainty, estimated by the spread of 1000 bootstrap model replications for each RCM).

At the 1.5 °C warming target, all regions exhibit a moderate increase in BA, with significant changes mostly in the Iberian Peninsula (Fig. 4, panels a and b). Larger increases in BA are foreseen for the +2 °C case (Fig. 4, panels c and d) with a larger number of eco-regions displaying significant changes. For the +3 °C case (Fig. 4, panels e and f), the BA shows even larger positive changes that are significant in the majority of eco-regions. The obtained BA increases are consistent with the SPEI projected changes, depicting an overall intensification of drought conditions across regions that increases progressively with the level of global warming (Supplementary Fig. 5), with drier conditions resulting from an increase in PET (Supplementary Fig. 6) and a general reduction in precipitation amount (spatially more heterogeneous than for PET; Supplementary Fig. 7), which is in line with previous studies, e.g.,[50,51].

The lower BA changes estimated by NSM than by SM are consistent with the hypothesis of an adjustment of the ecosystem structure. That is, if we consider only direct climate-fire linkages through the stationary model, the BA projections are higher (especially considering the 3 °C scenario) than if we consider also the potential indirect effects of climate-driven changes in fuel productivity. Non-stationary models reduce the sensitivity of fire activity to dry periods by taking into account potential changes in productivity as a result of warming. However, as already mentioned, several models can be used to fit Eq. 2, and therefore this result could be partially model-dependent. The sensitivity of the results to the details of the NSM model is provided in Fig. 5. The figure shows the spatially averaged changes for the +3 °C warming considering different climatic variables driving Eq. 2 (see Table 1). Several models result in larger changes and larger spread, making the model based on temperature a lower limit to the increase in BA. All scenarios indicate a large increase in BA at this warming level.

A common issue of statistical model predictions is that they make inferences based on extrapolation of the model outside its range of calibration. In our case, it is likely that for the hotter scenarios, temperature estimations exceed the range of historical values used for fitting the relationship of Eq. 2. While only 7 and 9 of the 40 eco-regions considered show temperatures that exceed the warmest historical values in the +1.5 and +2 °C future periods, for the +3 °C period, such regions are 16 (i.e., 40% of the domain). The exclusion of these areas has very little effect on the estimate of the impacts on BA, as shown in Fig. 6 (comparing

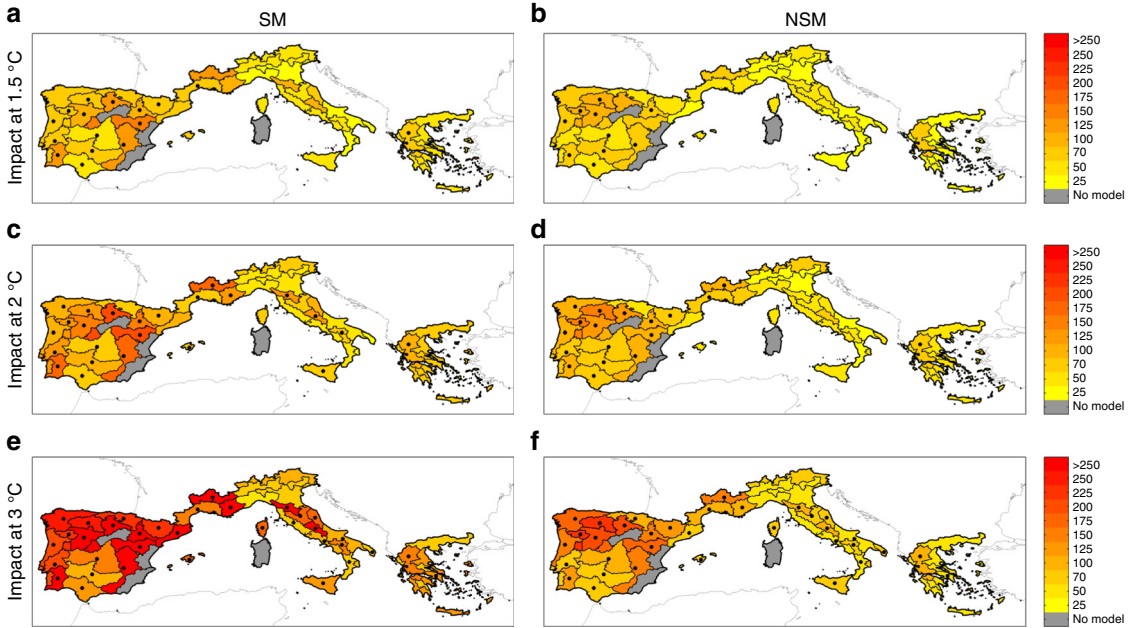

**Fig. 4** Ensemble mean burned area changes. Burned area changes (%) for **a** the +1.5 °C case with the stationary model SM (i.e., using Eq. 3), (**b**) the +1.5 °C case with non-stationary model NSM (i.e., NSM). using Eq. (4), (**c**) the +2 °C case with SM, (**d**) the +2 °C case with NSM, (**e**) the +3 °C case with SM, and **f** the +3 °C case with NSM. Dots indicate areas where at least 50% of the simulations (1000 bootstrap replications × the ensemble of RCMs) show a statistically significant change and more than 66% agree on the direction of the change. Coloured areas (without dots) indicate that changes are small compared to natural variations, and white regions (if any) indicate that no agreement between the simulations is found (similar to ref.[70])

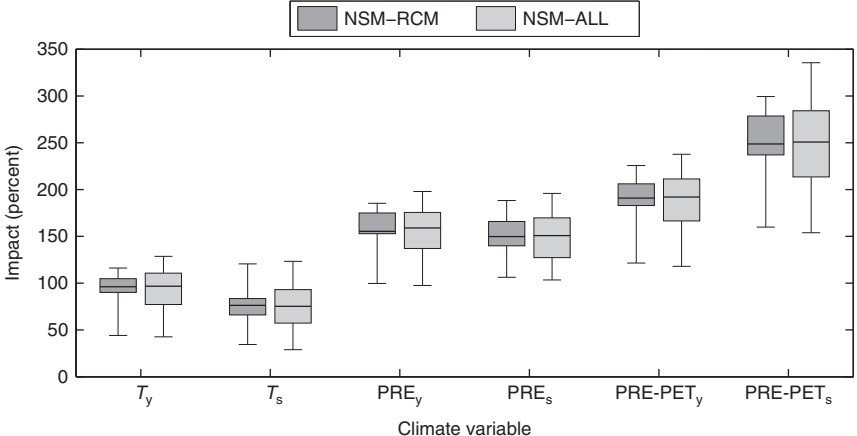

**Fig. 5** Sensitivity analysis of the burned area changes in Mediterranean Europe for +3 °C warming to the choice of the non-stationary model. Burned area changes (in %) are estimated considering the non-stationary model NSM (i.e., using Eq. 4) and several models with different climate variables (Table 1). Boxplots also show the uncertainty from the ensemble of RCM projections (NSM-RCM) and accounting for both RCM and regression model uncertainties (1000 bootstrap replications × the ensemble of RCMs; NSM-ALL). The median is shown as a solid line, the box indicates the 25–75 percentile range, while the whiskers show the 2.5–97.5 percentile range

the boxes with and without extrapolation), especially for the +1.5 and +2 °C scenarios. The future increase in BA, which is larger as the warming level increases, is thus confirmed. To further address the extrapolation issue, we also estimate BA changes when constraining $T_y$ and SPEI projections to historical extremes (following ref.[52]). In this case, we found no remarkable changes in our results, with only a slightly lower increase in BA. This result is presumably due to the partial compensation of two competing effects. On the one hand, constraining SPEI projections means that the BA values will be lower (following Eq. 1). On the other hand, constraining $T_y$ means that the climate-fire adjustments of Eq. 2, that led to lower BA changes, are also lower.

These results have been obtained considering bias-adjusted RCM data. Bias correction methods directly adjust the target variable projected by the climate model, using the corresponding local observations as references. One serious problem that may affect downscaling/bias correction methods is that they can modify the raw climate change signal (see, e.g.,[53–55]). The comparison between bias-corrected BA projections and the corresponding obtained with the direct RCM output (i.e., without bias correction) provide an estimation of the impact of the bias correction method in the results and, above all, allows us to assess whether the bias correction method preserves the climate change signal of the RCMs in the BA impacts. Although some differences appear, the main conclusions

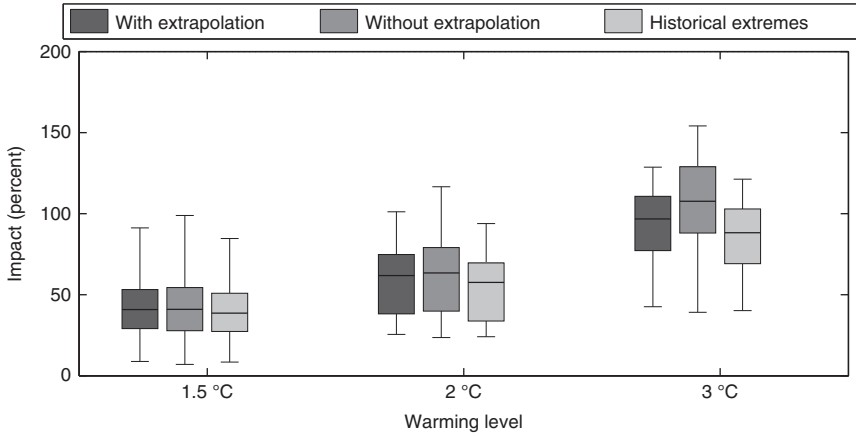

**Fig. 6** Sensitivity analysis of the burned area changes in Mediterranean Europe based on extrapolation. Burned area changes (in %) are shown for each warming level considering the non-stationary model NSM (i.e., using Eq. 4). Boxplots show the uncertainty from both RCM and regression model uncertainties (1000 bootstrap replications × the ensemble of RCMs) if predictors are allowed to exceed historical extremes (i.e., with extrapolation, as in the previous analyses), if eco-regions with temperature projections exceed the range of historical values used for fitting the relationship of Eq. 2 are excluded (i.e., without extrapolation) and if predictors are constrained to historical extremes. The median is shown as a solid line, the box indicates the 25–75 percentile range, while the whiskers show the 2.5–97.5 percentile range

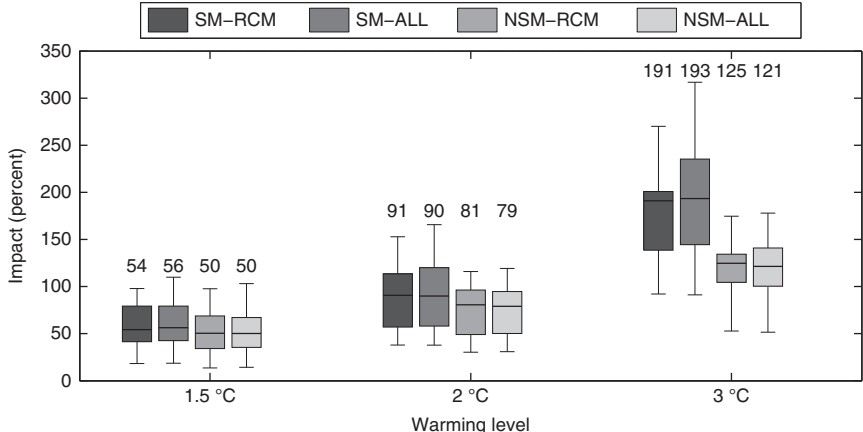

**Fig. 7** Burned area changes in Mediterranean Europe using climate data without bias correction. Same as Fig. 3 but with raw climate data, i.e., without bias correction

are confirmed also considering the direct RCM outputs, i.e., without performing any bias corrections (Fig. 7).

To summarise, our findings substantially align with the results of previous studies that assessed the impact of climate change on BA in the Mediterranean Basin, although comparisons are necessary limited since different scenarios, future periods and models are considered here. In ref.[22] the authors estimated increases of up to 66 and 140% in 2071–2100 relative to 1985–2004 under two IPCC scenarios, B2 and A2, respectively. In ref.[56], the authors obtained a 150–220% BA increase between 2000 and 2090 under A2 scenarios without considering adaptation, whereas they estimated a 74% increase in the adaptation scenario (prescribed burnings under present climate conditions). In ref.[25], the authors estimated 34% BA increases in Southern Europe, in 2070–2100 relative to 1960–1990 under the A1B scenario. These authors attributed such relatively low values to the projected human ignition/suppression probability and to the role of vegetation productivity. In ref.[26], the authors employed different fire-dynamic vegetation coupled models and increases of 14–17% and 60–71% were obtained between 1981–2000 and 2081–2100 periods under RCPs 2.6 and 8.5, respectively.

As a word of caution, we note that the methodology employed here has some limitations. Presumably, the complex relationships between climate, vegetation and fires hamper the applicability of fire impact models to conditions that are very different from the current ones. For these reasons, our estimate of fire response should be considered more robust for a few decades in the future, when climatic conditions should not be dramatically different from the current ones. Our model does not consider future changes in fire management policies, land-use and land-cover change, or in ignition patterns mainly because reliable projections for these drivers are not available. Future analyses could include the use of more complex drought metrics (e.g., that account for the response of plant transpiration induced by changing atmospheric $CO_2$ conditions[57]). A possible extension of the summer fire regime to previous months in spring and/or later months in autumn should also be explored. Despite these limitations, we illustrate that plausible levels of modification of the SPEI-BA links could reduce the impacts of climate change on BA, but still, at or above 1.5 °C of warming, Mediterranean BA is projected to increase, and at higher warming levels this increase becomes even larger. These results, in combination with the increase in societal exposure to large wildfires in recent years[58], call for a rethinking

of current management strategies[59]. Climate change effects could overcome fire prevention efforts, implying that more fire management efforts must be planned in the near future. The negative measured trend of BA in Mediterranean Europe in the past few decades can be explained by an increased effort in fire management and prevention[8]. However, keeping fire management actions at the current level might not be sufficient to balance a future increase in droughts. In this sense, the ability to model the link between climate and fire is crucial to identifying key actions in adaptation strategies. In particular, seasonal climate forecasts may enable a more effective and dynamic adaptation to climate variability and change, offering an under-exploited opportunity to reduce the fire impact of adverse climate conditions[60].

In summary, our results support the statement of the Paris Agreement that reports that limiting the temperature increase to 1.5 °C would "significantly reduce the risks and impacts of climate change".

## Methods

**Fire and drought data.** We obtained monthly BA (larger >1 ha) data from the EFFIS[61] dataset at the NUTS3 level (2006 version; see http://ec.europa.eu/eurostat/web/nuts/ for more details) for Portugal, Spain, southern France, Italy and Greece, for the period 1985–2011. We analysed the BA of the summer months from June to September. This is the period with large fires, which account for 86% of the annual BA. For more details on this dataset, see ref.[8].

We use the standard precipitation and evaporation index (SPEI)[43] to estimate drought intensity. SPEI uses as an input a water balance, taking into account the total accumulated precipitation and PET. The Hargreaves PET estimation method has been considered, taking into account temperature and precipitation in its formulation (plus the latitudinal correction factor). SPEI can be represented at different time scales and is thus able to effectively represent the multi-scalar aspect of droughts. Here, we consider three time scales, namely, 3, 6, and 12 months, as in refs.[9,62]. We calculated the observed SPEI for the period 1950–2015 from the publicly available gridded data set from the Climatic Research Unit of the University of East Anglia (CRU TS v4.0; 0.50 degree resolution[63]). PET and SPEI were calculated using the R package SPEI, version 1.7.

The future SPEI projections are calculated using two ensembles of nine regional climate simulations (involving four different RCMs and five GCM runs; Supplementary Table 2) spanning the period of 1970–2099. Both include the same members, but one assumes the moderate RCP4.5 and the other assumes the more extreme RCP8.5 scenario. The simulations were performed under the umbrella of the Euro-CORDEX project[64], covering Europe with a spatial resolution of 0.11 degrees both in latitude and longitude, the finest so far in this type of climatological multi-model and multi-scenario experiment. These are the RCMs that had the necessary variables at the moment of the design of our study, and they have been extensively validated (see, e.g.,[64,65]). The Euro-CORDEX data are interpolated to the regular 0.5 degree resolution grid of the CRU database using nearest-neighbours prior to any other further transformation. For each RCM, the parameters that are required to calculate the SPEI are determined relative to the distribution of the reference period 1971–2000 at each grid point. The fitted parameters are then used to calculate the historical and future SPEI series.

We bias corrected the RCM monthly climatic variables by applying a simple linear scaling applied at each grid point in the reference regular grid (CRU, 0.5 degree resolution). For PRE and PET, a scaling factor based on the ratio of the long-term mean (over the period 1971–2000) observed and simulated data are used as these are variables with a lower bound. For T, the difference between the long-term mean observed data and simulated data are used to correct the raw data. This is a simple and parsimonious bias correction method that intends to correct the mean bias. This method assumes that the bias is stationary in different climates and, correcting for the bias in the mean, corrects biases in the variance and quantiles of the distribution of the climatic variable. The raw SPEI projections have also been computed for benchmarking purposes.

The climate warming periods (1.5, 2, and 3 °C) are reported in Supplementary Table 3 and were selected for each simulation following the procedure described in ref.[34]. The time windows are defined as the earliest 30-year periods with time-averaged global mean temperature increase, as projected by the RCM-driving GCM simulation, equals to 1.5, 2, and 3 °C warming, respectively, compared to the 'pre-industrial' period 1881–1910. While for the 1.5 and 2 °C periods, we used the ensemble of nine simulations for each of the two RCPs (i.e., an ensemble of 18 members), for the 3 °C period, we consider only the simulations for the RCP8.5 scenario as this is the only scenario for which all the GCMs reach this warming within the study period.

The BA and the climatic data are then aggregated considering the 44 eco-regions defined by combining the available fire information with the environmental zones defined by ref.[66] (see Supplementary Fig. 4 and ref.[9] for more details).

Specifically, the SPEI data (observed and simulated) are calculated for each point of the 0.5 grid and then spatially averaged over these eco-regions.

**Drought-fire model development.** The procedure for developing the SPEI-BA model, following the work of ref.[9], includes the following steps. First, we normalise the positively skewed BA variables by applying a log transformation (i.e., $Y = \log(BA)$). Then, to identify the SPEI indicators, we (i) compute the correlation between $\log(BA)$ and $SPEI_{sc,m}$, with $sc = (3,6,12)$, $m = (0,7)$, i.e., summer and previous spring months; (ii) calculate the significance of the individual correlations (subject to the relationship between BA and SPEI being negative, i.e., a one-tailed hypothesis test). We estimate the correlation significance using bootstrap resampling, where one of the two variables is shuffled 1000 times and new correlations are computed. To account for the spatial dependence structure of the data, we use the same resampling sequence for all grid points within each bootstrap iteration (as in ref.[67]); (iii) we test the $p$-values of the previous step for multiple testing with a false discovery rate (FDR) test[68]; and (iv) we seek the minimum correlation values among all the significant correlations calculated in the previous steps. We also test for the presence of a relationship with the antecedent climate variable. However, no significant relationships have been found. When the BA or the SPEI time series exhibit a significant trend (i.e., $p$-value <0.05, assessed with the Mann–Kendall test), we fit the model including the predictor time $T$. When no trend is present, $\beta_3$ is set to zero. The regression coefficients are estimated using a robust regression procedure that adopts iteratively reweighted least squares with a bisquare weighting function[69]. We estimate the uncertainty of the parameters of the SPEI-BA model using bootstrap resampling, where the predictand and predictor pairs are drawn randomly with replacement 1000 times and new regression models are fit to the data. To account for the spatial dependence structure of the data, we use the same resampling sequence for all grid points within each bootstrap iteration.

The models are also assessed by means of a leave-one-out cross-validation, i.e., excluding the tested year when computing the model parameters.

We fit the model of Eq. 2 with and without considering the eco-region of the Alps (Supplementary Fig. 4), as this region seems to be an outlier (see the point with an averaged temperature below 5 °C in Fig. 2). Excluding this region, we obtain the model: $\beta_2 = -1.8 + 0.07\,Ty$ (bootstrapped 95% confidence intervals −2.2 to −1.4 for the intercept and 0.04 to 0.10 for the slope), which is similar to the model that includes this point, $\beta_2 = -1.6 + 0.06\,Ty$ (95% confidence intervals of −2.1 to −1.4 for the intercept term and of 0.04 to 0.09 for the slope).

**Code availability.** On behalf of reproducibility and applicability, the codes used in this work are available for research purposes by contacting the corresponding author. In any case the codes used for the data processing are mainly based on open source software: the Climate Data Operators (CDO version 1.7.2; functions: remapbil, remapcon) available from https://code.mpimet.mpg.de/projects/cdo and the R "Language and Environment for Statistical Computing" (R version 3.4.3) available from https://www.r-project.org/. Specifically, climate data access and processing has been undertaken using the open source R packages of the climate4R framework (http://www.meteo.unican.es/climate4R). A fully reproducible worked example of Euro-CORDEX data retrieval and calculation of observed and bias-corrected SPEI projections is provided online at http://www.meteo.unican.es/work/climate4r/drought4R/drought4R_notebook.html. The notebook source code is also available at https://github.com/SantanderMetGroup/notebooks. The climate–fire model development, the assessment of the climate and burned area projections, as well as their uncertainties, are mainly based on Matlab codes written by M.T. that are available for research purposes from the corresponding author upon request.

## Data availability

EFFIS data can be retrieved from the European Forest Fire Information System (http://forest.jrc.ec.europa.eu/effis/); Observed CRU data can be obtained from the University of East Anglia (https://crudata.uea.ac.uk/cru/data/hrg/). The EURO-CORDEX RCM models are publicly available through the Earth System Grid Federation infrastructure (ESGF, https://esgf.llnl.gov). In order to ensure the full reproducibility of the results, the authors will provide the data (observed and simulated) used in this study for research purposes to interested readers.

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

## Acknowledgements

We acknowledge the European Forest Fire Information System-EFFIS (http://effis.jrc.ec. europa.eu) of the European Commission Joint Research Centre for the fire data; the CRU data providers (https://crudata.uea.ac.uk/cru/data/hrg); the World Climate Research Programme's Working Group on Regional Climate; the Working Group on Coupled Modelling; the former coordinating body of CORDEX and panel responsible for CMIP5; the Earth System Grid Federation infrastructure, an international effort led by the U.S. Department of Energy's Program for Climate Model Diagnosis and Intercomparison; the European Network for Earth System Modelling; and other partners in the Global Organization for Earth System Science Portals (GO-ESSP). We also thank the climate modelling groups (listed in Supplementary Table 2) for producing and making available their model output. Participation of Antonello Provenzale is supported by the EU H2020 project 641762 "ECOPOTENTIAL" and the ERA-NET ERA4CS project"SERV-FOR-FIRE". Special thanks to Sergio M. Vicente Serrano, Esteve Canyameras, and Xavier Castro for their helpful discussions concerning the study. Marco Turco was supported by the Spanish Juan de la Cierva Programme (grant code: IJCI-2015-26953).

## Author contributions

M.T. conceived the study. M.T., J.J.R.C., J.B., S.J., and J.P.M. designed and carried out the data analysis and wrote the paper. M.C.L. and A.P. participated in the definition of the analysis methodology, contributed to discussing the results and participated in the writing of the text.

## Additional information

**Competing interests:** The authors declare no competing interests.

