## [Peer Review File · Nature Communications]

Reviewer #1 (Remarks to the Author):

This paper seeks to estimate changes in area burned in the Mediterranean region of Europe under three warming scenarios, 1.5, 2 and 3 C. Though several papers have made projections of future areas burned under warming scenarios, this paper is novel in its attempt to address the non-linear effects of changing vegetation structure on area burned as temperatures warm. There is some concern that (log) linear models of temperature vs area burn over-predict fire under future warming, due to assumptions about static vegetation/fuels, so the idea is of interest to a wide community of scientists, managers and the general public, both in the Mediterranean region but also to other fire-prone regions such as California and Australia. I was pleased to see the authors attempt to incorporate change in vegetation structure into their projections of future burn area under a variety of warming scenarios. However, as it stands, the solution violates at least two statistical assumptions. First, SPEI and PET are calculated from the same variables, so using them both simultaneously in a regression setting is not appropriate. Further, PET would only loosely be associated with the actual on the ground vegetation structure which likely has been affected by a variety of land uses. Sure, the regression (Fig 2) explains ~36% of the variability in beta-2, but PET is a stand in for some other variable and has the problem of being correlated directly with SPEI. Finally, I had some concerns about the spatial autocorrelation in the AB statistics as well as the predictors (though this is less problematic). A more elegant solution, that would address all of these issues would be to use actual fuel structure or vegetation structure data from each ecoregion (for example satellite-derived) in a spatial regression setting. One model could take care of the whole gig.

I would also like to note that there has been some discussion among climate modelers about the reliability of drought indices calculated post-hoc from climate model output. Though I do not think this item in itself should prevent publication, the authors may want to be aware of this discussion if they are not already. There are further plant responses to elevated CO₂ that could ameliorate drought that may make direct calculations from the model output inaccurate, for example:
Plant responses to CO₂ reduce estimates of drought
Abigail L. S. Swann, Forrest M. Hoffman, Charles D. Koven, James T. Randerson
Proceedings of the National Academy of Sciences Sep 2016, 113 (36) 10019-10024;
DOI:10.1073/pnas.1604581113

One final comment: though I try not to be influenced by grammatical details, especially when the authors are from a non-English speaking country, there were some sections that were difficult to read and interpret. It may be worth passing this by an English editor before resubmitting.

Specific Questions:

L.285 A simple local scaling approach has been used in order to correct the biases of the simulated data.

Please define how this scaling was performed. Was it based on area or ?

L/296-299. Not clear what is meant here. Perhaps strike these lines?

Reviewer #2 (Remarks to the Author):

The manuscript entitled "Impact of climate change on summer fires in Mediterranean Europe at 1.5, 2 and 3°C warming" by Turco et al. examines future burnt areas (BA) under different climatic scenarios (1.5, 2 and 3°C warming) over the euro-Mediterranean area. They built regressions-scale regressions between summer BA and the Standardized Precipitation Evaporation Index (SPEI) and then projected these relationships for different climate scenarios with and without

taking into account how the long-term impact of climate on vegetation might affect the fire-climate relationships. They found that limiting the temperature increase below the 1.5 °C level would lead to double the BA in the future with almost no effect of long-term vegetation changes. Above this 1.5 °C threshold BA increases sharply for 3°C scenarios the BA increase is about 200 % but is considerably reduced (130 %) when taking into account the impact of long-term effects on vegetation and its impact on the fire-climate relationship.

This is a very interesting study. Overall, the manuscript is well-written and the language is clear. The methodology that uses regression analysis between SPEI and BA is both straightforward and relevant, and draws upon an important paper of the same author published last year in Scientific Reports. The analysis and applications of climate change scenarios for BA projections sounds also robust although more details could be provided in order to clarify the methodological choices and make their analysis more easily reproduced (see some of my minor comments below), but this is mostly of minor importance.

One of the most important findings of this study is that temperature increase should be limited to 1.5°C to avoid some potential large increase in BA and that even with this best scenario, BA could be multiplied by a factor 2. This claim is novel, convincing and will be of great value for the field as well as for the geoscience community in general. I also believe that the maps of future BA are likely to be a new major reference for a large number of studies. However, I found that results could be more discussed in the context of previous literature. The authors could provide more detailed comparisons of these new simulations with previous studies that use different methodologies either at similar scales or at more local-scale (country or region). Similarly, I think that these results should be compared with the uncertainty resulting from other non-climatic factors, e.g. what is the meaning of a 50 % increase in BA compared to the suppression related decrease observed in the last decades in the euro-Mediterranean. I think that a more thorough discussion about these points will likely strengthen the results of this study without requiring additional analyses.

One main originality of this study is to assess how long-term impacts of changing climate on vegetation structure and/or composition affect the fire-weather relationships and future BA estimates. The authors claim this effect is only significant for the scenarios that led to the largest increase in BA. Although the methodology proposed to take into account this effect is rather elegant and takes advantage of the important drought gradient observed in the Mediterranean, I have however a number of concerns regarding this analysis. First, my opinion is that the hypothesis about the non-stationarities in the fire-climate relationships is not appropriately introduced. Examples of studies that specifically address non-stationarities in fire-weather relationships are numerous in recent literature but this theoretical framework is not mentioned. (e.g. Pausas & Paula 2012, Higuera et al. 2015, Ruffault & Mouillot 2015, Erni et al. 2017, Keeley & Syphard 2017, Syphard et al. 2017). The authors should rely on the results and frameworks of these studies to clearly state their working hypothesis and discuss their results. Second, it is likely that for the drier scenarios, ETP estimations are beyond the range of ETP used for fitting the relationship between BA and ETP (with more than 50 % increase in some already dry areas Fig. S8). Yet, it is acknowledged that the relationship between drought and BA is not similar whether we consider a drought or a fuel limited ecosystem. This limitation questions the validity of your relationship outside its range of calibration that you should discuss in the context of your analysis.

Julien Ruffault

Other comments:

- L22-23: This sentence is not clear. Please provide some details about the non-stationarities you are dealing with in this paper. Similar comment applies to L30-31
- L22-23: non-stationarities?
- L58: That's an interesting question but you did not answer this "2017 fire season issue" in your paper.
- L70: summer "drought conditions"
- L77-82: This hypothesis should be better introduced.
- L89-94:

- L168-L172. This should be mentioned earlier in the manuscript.
- L154-158: I'm wondering why not test for the correlation between 2 and an aridity index including both PPT and ETP in a single metric (like SPEI)?
- L165-168: I am not sure I understand your conclusion here. What do you mean by "vegetation is better adapted to water scarcity"?
- L198-199: What is the purpose of compiling model outputs without bias corrections ?
- L200: Please provide some details to explain while these results are consistent with the SPEI changes.
- L193-194: This part is described in the M&M section, not in sup mat.
- L223-226: This seems overstated as you did not explicitly test this effect. I suggest to analyze the importance (in %) of each effect (ETP and PPT SPEI and long term ETP though its impact on 2) on BA increase.
- L217-229; I don't think this paragraph that essentially summarized the study is necessary.
- L232-234. This point is arguable. For instance, there is still a lot of uncertainty associated to rainfall projections. In addition, I think this point needs to be lengthened to include more references and differentiate the uncertainty related to projections themselves from the uncertainty related to the effect of these factors on the fire weather relationship.
- L237: Why an extension of the fire season only towards autumn and not spring?
- L255: large not "larger"
- L280-281: what motivated the choice of these specific ESM-RCM couplings?
- L280-288. I suppose SPEI values for the future period have been determined relative to the distribution of the reference period?
- L284: The bias-correction method is not clear to me. What is the "local scaling approach" here and what are the assumptions associated to this method?
- References to supplementary figures are wrong in many places. Please carefully check this throughout the manuscript.

Figures

- Figure 2 is nice but one case (around 450 mm PET) seems to be very influential for the regression line and I suspect that the results will be much different if you exclude it from analyses. Have you checked the leverage effect on for this regression? You might also consider non-parametric fitting to reduce the weight of this case. Also, I suppose the authors have already thought about this point but I wonder if the nature of the relationship might change without this point (logarithmic). In any case, you should provide confidence intervals.

References

- Erni, S., Arseneault, D., Parisien, M. A., & Bégin, Y. (2017). Spatial and temporal dimensions of fire activity in the fire-prone eastern Canadian taiga. *Global change biology*, 23(3), 1152-1166.
- Higuera, P. E., Abatzoglou, J. T., Littell, J. S., & Morgan, P. (2015). The changing strength and nature of fire-climate relationships in the northern Rocky Mountains, USA, 1902-2008. *PloS one*, 10(6), e0127563.
- Keeley, J. E., & Syphard, A. D. (2017). Different historical fire-climate patterns in California. *International Journal of Wildland Fire*, 26(4), 253-268.
- Pausas, J. G., & Paula, S. (2012). Fuel shapes the fire-climate relationship: evidence from Mediterranean ecosystems. *Global Ecology and Biogeography*, 21(11), 1074-1082.
- Ruffault, J., & Mouillot, F. (2015). How a new fire-suppression policy can abruptly reshape the fire-weather relationship. *Ecosphere*, 6(10), 1-19.
- Syphard, A. D., Keeley, J. E., Pfaff, A. H., & Ferschweiler, K. (2017). Human presence diminishes the importance of climate in driving fire activity across the United States. *Proceedings of the National Academy of Sciences*, 201713885.

Reviewer #3 (Remarks to the Author):

Manuscript Review "Impact of climate change on summer fires in Mediterranean Europe at 1.5, 2 and 3°C warming" by Turco and collaborators

Technical and editorial comments are detailed below. Many editorial comments were written directly on the manuscript. The references were not scanned for errors.

GENERAL COMMENTS

This study presented interesting results on the projection of fire activity into the future under various climate warming scenarios. I really like that the authors decided NOT to use standard IPCC RCP scenarios, but rather used scenarios of their own choosing, which puts the control in their hands. Unfortunately, this is such a poorly written and organized paper that it really can't be published at this time. Here are some major problems I found in the paper:

1. Poorly written. Many concepts, techniques, and statements are not clearly presented. Many terms and not cited with the appropriate literature.
2. Poorly cited. There are literally thousands of papers on this topic for Australia and North America – where were those papers?
3. Wordy. The entire Results section could have been reduced by 40% by concentrating on the important messages.
4. Organize. Why are there methods in the Results section?

One main concern in this manuscript is that it seems that the results don't tell us anything we don't already know, and the presentation of the results don't live up to the title and abstract.

GENERAL COMMENTS

1. Title –How about getting rid of the "at 1.5, 2., and 3deg warming"?
2. Abstract – Fails to capture the essence of the study and its implications
3. First sentence is totally wrong – these are not the main factors, especially soil moisture – they are only some of the factors. Weather is identified as the most important factor.
4. Organization. I really didn't like the fact that Results were meshed with Discussion. The authors seemed to take liberties with space – much of the written text seems redundant or unneeded. In fact, the entire section could be reduced and reorganized to improve understanding. Some of the discussion material could be included in the introduction, for example.
5. Terminology. Many terms throughout the document were not defined or cited.
6. Figures and Tables
 - a. I think many of the figures could have been combined and some figures could be removed. The authors should concentrate on the figures that are meaningful to the results, not to understanding the methods.
 - b. Many of the tables need modifications as shown in the marked MS

TECHNICAL COMMENTS

1. Methods
 - a. Why are there no methods in this paper? It looks like the methods are in the Results?
 - b. Equation 1 is badly described. Variables undefined, mixed scales.
 - c. Having some methods at the end is very confusing.
 - d. This section could be reduced.

Responses to reviewer #1

Reviewer #1 (Highlight): *This paper seeks to estimate changes in area burned in the Mediterranean region of Europe under three warming scenarios, 1.5, 2 and 3 C. Though several papers have made projections of future areas burned under warming scenarios, this paper is novel in its attempt to address the non-linear effects of changing vegetation structure on area burned as temperatures warm.*

Comments

1. Referee's Comment: *There is some concern that (log) linear models of temperature vs area burn over-predict fire under future warming, due to assumptions about static vegetation/fuels, so the idea is of interest to a wide community of scientists, managers and the general public, both in the Mediterranean region but also to other fire-prone regions such as California and Australia. I was pleased to see the authors attempt to incorporate change in vegetation structure into their projections of future burn area under a variety of warming scenarios. However, as it stands, the solution violates at least two statistical assumptions. First,*

SPEI and PET are calculated from the same variables, so using them both simultaneously in a regression setting is not appropriate.

Response: Similar evaluations of the risk of over or under-prediction under future warming due to assumptions about static climate impact models have been undertaken in other potentially vulnerable sectors. For instance, in agriculture, specific methods have been developed to estimate the adaptation potential in response to climate change (see e.g. Butler and Huybers, 2013; Moore and Lobell, 2014). Our approach follows the study of Butler and Huybers (2013) to explore the potential non-stationary response of fires to climate under changed conditions. In essence, this method first develops an ensemble of time-series regression models for each region, linking predictand (BA) and predictor (SPEI, Eq. 1) and then uses cross-sectional regression (Eq. 2) to analyse the sensitivity of the fire-climate relationships to long-term averaged climate indicators across all regions. We then use such a spatial variation, identified with Eq. 2, as a proxy for the modifications of the fire-climate relationships, identified with Eq. 1, under changed climatic conditions (i.e. adjusted to changes in the long-term averaged climate indicators used in Eq. 2). Thus, Eq. 1 considers variables to be linked with a sequence of points in time, while in Eq. 2 both predictand and predictor variables refer to one specific period in time (i.e. the average over the calibration period). We used SPEI for time series models (Eq. 1) and the annual PET long-term average for the cross-sectional model (Eq. 2), although, for the reasons explained below, we replaced PET by temperature (T) in the cross-sectional model in the revised manuscript. Thus, this approach does not consider a multi-variable regression simultaneously including SPEI and PET (or SPEI and T), so the danger of over-fitting is not present. Indeed, we could have used the same predictor variable in both models (Eqs. 1 and 2), but SPEI has a zero-mean by definition in each grid point in the calibration (i.e. present) period, which hampers its use in the model of Eq. 2. To make this clearer, while previously the dependence on time and space of the variables used in Eqs. 1 and 2 was omitted for ease of notation, we have now modified the nomenclature of the equations in the revised manuscript.

2. Referee's Comment: *Further, PET would only loosely be associated with the actual on the ground vegetation structure which likely has been affected by a variety of land uses. Sure, the regression (Fig 2) explains ~36% of the variability in beta-2, but PET is a stand in for some other variable and has the problem of being correlated directly with SPEI. Finally, I had some concerns about the spatial autocorrelation in the AB statistics as well as the predictors (though this is less problematic). A more elegant solution, that would address all of these issues would be to use actual fuel structure or vegetation structure data from each ecoregion (for example satellite-derived) in a spatial regression setting. One model could take care of the whole gig.*

Response: We agree that the reviewer's proposal to use vegetation data to construct the model of Eq. 2 could help to explain the spatial variations in the fire-climate relationships identified with Eq. 1. We tested it. Nonetheless, the model of Eq. 2 constructed with satellite-derived vegetation data (Normalized Difference Vegetation Index, NDVI data; see Zhu et al., 2013) explained at best only ~21% of the variability in beta-2, i.e. using NDVI data aggregated at summer scale (the NDVI data aggregated at annual scale did not show a statistically significant correlation). This is a lower percentage than with the models that instead consider climatic variables data (see Table 1 in the revised manuscript). Besides, such a model (Eq. 2 using NDVI data) would not be valid for exploring the non-stationary response of BA to climate under changed conditions because no future projections for the NDVI exist.

We have addressed the autocorrelation issue regarding the model of Eq. 2 by using Moran's I test for spatial autocorrelation of the residuals. This test revealed that the residuals of the models that consider the PET variable as predictor (both at annual or summer scales) are spatially correlated, and for this reason, the PET-based models have now been discarded. We also tested whether the inclusion of more than one predictor (i.e. linear combinations of PET, T and PRE) improves the model of Eq. 2. However, these climate variables are spatially correlated, hampering a multi-variable model development due to the danger of over-fitting. Thus, we retained only the variables T, PRE and the PRE-PET water balance indicator (this latter was suggested by reviewer#2) as candidates as predictors in the model of Eq. 2. Finally, the best option by which the hypothesis of negligible spatial autocorrelation of the residuals is satisfied, is the model based on the long-term mean temperature. However, the other potential models (with summer temperature, PRE or PRE-PET) also showed skill in reproducing beta-2 (with significant correlations between simulated and observed values, see Table 1 in the revised manuscript). Since model selection is a critical step, in the revised paper we tested the sensitivity of the results to the choice of model (see Figure 5 in the revised manuscript).

3. Referee's Comment: *I would also like to note that there has been some discussion among climate modelers about the reliability of drought indices calculated post-hoc from climate model output. Though I do not think this item in itself should prevent publication, the authors may want to be aware of this discussion if they are not already. There are further plant responses to elevated CO2 that could ameliorate drought that may make direct calculations from the model output inaccurate, for example:*

Plant responses to CO2 reduce estimates of drought

Abigail L. S. Swann, Forrest M. Hoffman, Charles D. Koven, James T. Randerson

Proceedings of the National Academy of Sciences Sep 2016, 113 (36) 10019-10024; DOI:10.1073/pnas.1604581113

Response: This discussion is now properly included in the manuscript (see lines 295-297).

4. Referee's Comment: *One final comment: though I try not to be influenced by grammatical details, especially when the authors are from a non-English speaking country,*

there were some sections that were difficult to read and interpret. It may be worth passing this by an English editor before resubmitting.

Response: The Nature Research Editing Service has professionally proofread the manuscript.

Specific Questions:

1. Referee's question: *L.285 A simple local scaling approach has been used in order to correct the biases of the simulated data.*

Please define how this scaling was performed. Was it based on area or ?

Response: Bias correction is performed at the grid box level. For PRE and PET, a scaling factor based on the ratio of the long-term mean observed and simulated data is used. For T, the difference of the long-term mean observed data and simulated data is used to scale or correct the raw data. This procedure follows Teutschbein & Seibert (2012). We now include more details on the way we bias correct the data.

2. Referee's question: *L/296-299. Not clear what is meant here. Perhaps strike these lines?*

Response: We have rewritten them, see lines 353-357.

References

- Butler, E. E. & Huybers, P. Adaptation of us maize to temperature variations. *Nature Climate Change* 3, 68–72 (2013).
- Moore, F. C. & Lobell, D. B. Adaptation potential of European agriculture in response to climate change. *Nature Climate Change* 4, 610–614 (2014).
- Teutschbein, C., & Seibert, J. Bias correction of regional climate model simulations for hydrological climate-change impact studies: Review and evaluation of different methods. *Journal of Hydrology*, 456, 12-29 (2012).
- Zhu, Z., et al. Global data sets of vegetation leaf area index (LAI) 3g and fraction of photosynthetically active radiation (FPAR) 3g derived from global inventory modeling and mapping studies (GIMMS) normalized difference vegetation index (NDVI3g) for the period 1981 to 2011. *Remote sensing*, 5(2), 927-948 (2013).

Responses to reviewer #2

Reviewer #2 (Highlight): *The manuscript entitled “Impact of climate change on summer fires in Mediterranean Europe at 1.5, 2 and 3°C warming” by Turco et al. examines future burnt areas (BA) under different climatic scenarios (1.5, 2 and 3°C warming) over the euro-Mediterranean area. They built regressions-scale regressions between summer BA and the Standardized Precipitation Evaporation Index (SPEI) and then projected these relationships for different climate scenarios with and without taking into account how the long-term impact of climate on vegetation might affect the fire climate relationships. They found that limiting the temperature increase below the 1.5 °C level would lead to double the BA in the future with almost no effect of long-term vegetation changes. Above this 1.5 °C threshold BA increases sharply for 3°C scenarios the BA increase is about 200 % but is considerably reduced (130 %) when taking into account the impact of long-term effects on vegetation and its impact on the fire-climate relationship.*

This is a very interesting study. Overall, the manuscript is well-written and the language is clear. The methodology that uses regression analysis between SPEI and BA is both straightforward and relevant, and draws upon an important paper of the same author published last year in Scientific Reports. The analysis and applications of climate change scenarios for BA projections sounds also robust although more details could be provided in order to clarify the methodological choices and make their analysis more easily reproduced (see some of my minor comments below), but this is mostly of minor importance.

Response: We have done our best to improve the manuscript following the reviewer's recommendations. Below we detail the modifications made to address each point.

Comments

1. Referee's Comment: *One of the most important finding of this study is that temperature increase should be limited to 1.5°C to avoid some potential large increase in BA and that even with this best scenario, BA could be multiplied by a factor 2. This claim is novel, convincing and will be of great for the field as well as for the geoscience community in general. I also believe that the maps of future BA are likely to be a new major reference for a large number of studies. However, I found that results could be more discussed in the context of previous literature. The authors could provide more detailed comparisons of these new simulations with previous studies that use different methodologies either at similar scales or at more local scale (country or region). Similarly, I think that these results should be compared with the uncertainty resulting from others non-climatic factors, e.g. what is the meaning of a 50 % increase in BA compared to the suppression related decrease observed in the last decades in the euro-Mediterranean. I think that a more thorough discussion about these points will likely strengthen the results of this study without requiring additional analyses.*

Response: New references have been added to better contextualize our findings (see lines 268-287), as well as a deeper discussion on the uncertainty/impacts resulting from other non-climatic factors (see lines 288-298 and 303-308).

2. Referee's Comment: *One main originality of this study it to assess how long-term impacts of changing climate on vegetation structure and/or composition affect the fire-*

weather relationships and future BA estimates. The authors claim this effect is only significant for the scenarios that led to the largest increase in BA. Although the methodology proposed to take into account this effect is rather elegant and takes advantage of the important drought gradient observed in the Mediterranean, I have however a number of concerns regarding this analysis. First, My opinion is that the hypothesis about the nonstationarities in the fire-climate relationships is not appropriately introduced. Examples of studies that specifically address non-stationarities in fire-weather relationships are numerous in recent literature but this theoretical framework is not mentioned. (e.g. Pausas & Paula 2012, Higuera et al. 2015, Ruffault & Mouillot 2015, Erni et al. 2017, Keeley & Syphard 2017, Syphard et al. 2017). The authors should rely on the results and frameworks of these studies to clearly state their working hypothesis and discuss their results.

Response: We have now included a more in-depth description of the hypothesis about the non-stationarities in the fire-climate relationships based on the existing literature (see lines 65-83), as suggested by the reviewer, as well as adding a discussion of our findings on the basis of these previous works (see lines 177-179).

3. Referee's Comment: *Second, it is likely that for the drier scenarios, ETP estimations are beyond the range of ETP used for fitting the relationship between 2 and ETP (with more than 50 % increase in some already dry areas Fig. S8). Yet, it is acknowledged that the relationship between drought and BA is not similar whether we consider a drought or a fuel limited ecosystem. This limitation questions the validity of your relationship outside its range of calibration that you should discuss in the context of your analysis.*

Response: We now include additional analyses on the sensitivity of our inferences based on extrapolation. Although we found small differences in the main results, it was an interesting exercise and is included in the main manuscript (see details in lines 243-254 and Figure 6).

Other comments:

1. Referee's question: *L22-23: This sentence is not clear. Please provide some details about the non-stationarities you are dealing with in this paper. Similar comment applies to L30-31*

L22-23: non-stationarities?

Response: We have revised the entire abstract and the term 'non-stationarities' is now properly contextualized and introduced.

2. Referee's question: *L58: That's an interesting question but you did not answer this "2017 fire season issue" in your paper.*

Response: The reviewer is right. We have removed the question.

3. Referee's question: *L70: summer "drought conditions"*

Response: Added.

4. Referee's question: *L77-82: This hypothesis should be better introduced.*

Response: The hypothesis about the non-stationarities in the fire-climate relationships is now better contextualized, introduced and discussed in the Introduction section.

5. Referee's question: *L168-L172. This should be mentioned earlier in the manuscript.*

Response: Done.

6. Referee's question: *L154-158: I'm wondering why not test for the correlation between 2 and an aridity index including both PPT and ETP in a single metric (like SPEI)?*

Response: We repeated the analysis (trained and tested the model in Eq. 2) considering the PRE-PET water balance indicator in addition to the long-term mean temperature (T), PRE and PET, considering the data aggregated both at annual and seasonal (summer) scales. The results are provided in Table 1. It can be seen that this indicator does not improve the models feed with PRE or T individually.

7. Referee's question: *L165-168: I am not sure I understand your conclusion here. What do you mean by "vegetation is better adapted to water scarcity"?*

Response: The sentence was certainly too brief to be understandable. It has been removed and the whole Results and discussion section reorganized, following the reviewers' recommendations.

8. Referee's question: *L198-199: What is the purpose of compiling model outputs without bias corrections ?*

Response: Regional Climate Models explicitly solve mesoscale atmospheric processes and provide spatially and physically consistent outputs. However, they still have considerable biases (see e.g. Turco et al., 2013; Kotlarski et al., 2014) which are typically adjusted in practical applications using a variety of Model Output Statistics (MOS) methods (Maraun et al., 2010). However, downscaling/MOS methods may have important drawbacks and best practice has not been established yet (Maraun et al., 2017; Jerez et al., 2018). One serious problem that may affect MOS methods is that they can modify the raw climate change signal (see e.g. Maurer and Pierce, 2014; Turco et al., 2017). That is, although bias correction is generally considered a necessary step in climate change impact studies, it introduces a further source of uncertainty (Maraun et al., 2017), possibly destroying the physically-based relationships between different climatic variables. The comparison between bias-corrected BA projections and the corresponding projections obtained with the direct RCM output (i.e. without bias correction) provides an estimation of the impact of the bias correction method on the results and, above all, allows us to assess whether or not the bias correction method adopted preserves the climate change signal of the RCMs in analysing future impacts on BA. It does, indeed, preserve it, and we show that both approaches produce similar results, as the revised manuscript now acknowledges.

9. Referee's question: *L200: Please provide some details to explain while these results are consistent with the SPEI changes.*

Response: We changed this sentence to:

"The obtained BA increases are consistent with the SPEI projected changes, depicting an overall intensification of drought conditions across regions that increases progressively with the level of global warming (Supplementary Fig. 5) [...]"

10. Referee's question: *L193-194: This part is described in the M&M section, not in sup mat.*

Response: We describe the procedure to select the future periods in the Methods section, but the periods in which the various warming levels are reached in each GCM simulation are provided in Supplementary Table 3.

11. Referee's question: *L223-226: This seems overstated as you did no explicitly test this effect. I suggest to analyze the importance (in %) of each effect (ETP and PPT SPEI and long term ETP though its impact on 2) on BA increase.*

Response: Although we find the reviewer's suggestion interesting, such an analysis would be by no means straightforward and would fall outside the scope of the present study. As we conceived the BA-climate model (Eq. 1), based on the existing literature and our own experience, it uses the SPEI as climate predictor for BA. PET and PRE are combined in this single index in a complex way that does not allow the relative contribution of these two variables to be separated. Specifically, we calculate the SPEI index as recommended by Beguería et al. (2014), thus considering a log-Logistic distribution function for computing the SPEI and the unbiased Probability Weighted Moment method for model fitting.

Finally, at L223-226, we were commenting on the fact that PET projections depicted much more intense patterns than those for PRE, but it was certainly incorrect to assert that the increase in drought conditions is mainly due to the projected increase in PET based on that alone. These lines have been removed, as suggested by the reviewer's in the comment below.

12. Referee's question: *L217-229; I don't think this paragraph that essentially summarized the study is necessary.*

Response: These lines have been eliminated in the revised paper.

13. Referee's question: *L232-234. This point is arguable. For instance, there is still a lot of uncertainty associated to rainfall projections. In addition, I think this point needs to be lengthened to include more references and differentiate the uncertainty related to projections themselves from the uncertainty related to the effect of these factors on the fire weather relationship.*

Response: We concur and accordingly introduced further analysis of the uncertainty associated with various sources. First, we estimated the RCM model uncertainty considering the spread of the ensemble of RCM projections. We then estimated the uncertainty associated with the model parameters when constructing the climate-fire models (Eqs. 1 and 2) considering the spread of 1000 model bootstrap replications. We also evaluated the impact of the choice of the predictor for the model of Eq. 2 to estimate the potential non-stationary response of BA to climate change, of the extrapolation of present-day relationships into the future (since future climate variables may exceed the historical extremes used to develop the empirical models), and of the use of bias-corrected or raw RCM data. We thus extended the Results and discussion section considerably, adding new figures (2 to 7, some of which are not exactly new figures but modified, enriched ones) and a deeper discussion on the robustness of our results. Finally, following the reviewer's recommendation, we rewrote L232-234 when discussing the availability of reliable projections for non-climatic factors (e.g. ignition patterns, fire management policies).

14. Referee's question: *L237: Why an extension of the fire season only towards autumn and not spring?*

Response: We changed this line to:

“to previous months in spring and/or later months in autumn”.

15. Referee's question: *L255: large not “larger”*

Response: Corrected.

16. Referee's question: *L280-281: what motivated the choice of these specific ESM-RCM couplings?*

Response: The motivation was the data availability at the time this study was performed, as acknowledged in the revised manuscript. Nonetheless, the GCM-RCM pairs chosen cover a wide range of the full GCM-RCM matrix of the Euro-CORDEX initiative: 5 out of 12 different

ESMs were used coupled to 4 out of 10 RCMs. The ensemble used in this study should provide a good overview of the model variability, so we deemed it appropriate to support our conclusions.

17. Referee's question: *L280-288. I suppose SPEI values for the future period have been determined relative to the distribution of the reference period?*

Response: Yes, we have added the following explanation in the revised paper:

“For each RCM, the parameters that are required to calculate the SPEI are determined relative to the distribution of the reference period 1971-2000 at each grid point. The fitted parameters are then used to calculate the historical and future SPEI series”.

18. Referee's question: *L284: The bias-correction method is not clear to me. What is the “local scaling approach” here and what are the assumptions associated to this method?*

Response: Bias correction is performed at the grid box level. For PRE and PET, a scaling factor based on the ratio of long-term mean observed and simulated data is used. For T, the difference between the long-term mean observed data and simulated data is used to scale or correct the raw data. This procedure follows Teutschbein & Seibert (2012). We now include more details on the way we bias correct the data and on the implicit assumptions associated with these corrections (see lines 344-351).

19. Referee's question: *References to supplementary figures are wrong in many places. Please carefully check this throughout the manuscript.*

Response: Done.

20. Referee's question: *Figures:*

Figure 2 is nice but one case (around 450 mm PET) seems to be very influential for the regression line and I suspect that the results will be much different if you exclude it from analyses. Have you checked the leverage effect on for this regression? You might also consider nonparametric fitting to reduce the weight of this case. Also, I suppose the authors have already thought about this point but I wonder if the nature of the relationship might change without this point (logarithmic). In any case, you should provide confidence intervals.

Response: The regression coefficients of Eqs. 1 and 2 are now estimated using a robust regression procedure that adopts iteratively reweighted least squares with a bisquare weighting function (Street et al., 1988). Such an approach is less sensitive to outliers than the classic least-squares estimator. We also estimate the confidence interval for the model parameters of Eqs. 1 and 2 using 1000 bootstrap resampling. These uncertainty ranges are now shown in Figure 2. The model of Eq. 2 now reads: $\beta_2 = -1.6 + 0.06 T_{\text{mean}}$ (where T_{mean} is the mean temperature climatology), with bootstrapped 95% confidence intervals (c.i.) of -2.1 to -1.4 for the intercept term and 0.04 to 0.09 for the slope. The 'outlier' point, with an averaged temperature below 5 degrees Celsius, corresponds to the Alpine region. Excluding this point, we obtain the model $\beta_2 = -1.8 + 0.07 T_{\text{mean}}$ (c.i. -2.2 - -1.4 for the intercept; 0.04 to 0.10 for the slope), which is quite similar to the previous model.

References

Beguiría, S., Vicente-Serrano, S. M., Reig, F., & Latorre, B. (2014). Standardized precipitation evapotranspiration index (SPEI) revisited: parameter fitting, evapotranspiration models, tools, datasets and drought monitoring. *International Journal of Climatology*, 34(10), 3001-3023.

- Jerez, S., López-Romero, J. M., Turco, M., Jiménez-Guerrero, P., Vautard, R., & Montávez, J. P. (2018). Impact of evolving greenhouse gas forcing on the warming signal in regional climate model experiments. *Nature communications*, 9(1), 1304.
- Kotlarski, S., Keuler, K., Christensen, O. B., Colette, A., Déqué, M., Gobiet, A., ... & Nikulin, G. (2014). Regional climate modeling on European scales: a joint standard evaluation of the EURO-CORDEX RCM ensemble. *Geoscientific Model Development*, 7(4), 1297-1333.
- Maraun, D., Shepherd, T. G., Widmann, M., Zappa, G., Walton, D., Gutiérrez, J. M., ... & Mearns, L. O. (2017). Towards process-informed bias correction of climate change simulations. *Nature Climate Change*, 7(11), 764.
- Maraun, D., Wetterhall, F., Ireson, A. M., Chandler, R. E., Kendon, E. J., Widmann, M., ... & Venema, V. K. C. (2010). Precipitation downscaling under climate change: Recent developments to bridge the gap between dynamical models and the end user. *Reviews of Geophysics*, 48(3).
- Maurer, E. P., & Pierce, D. W. (2014). Bias correction can modify climate model simulated precipitation changes without adverse effect on the ensemble mean. *Hydrology and Earth System Sciences*, 18(3), 915-925.
- Street, J. O., Carroll, R. J., & Ruppert, D. (1988). A note on computing robust regression estimates via iteratively reweighted least squares. *The American Statistician*, 42(2), 152-154.
- Teutschbein, C., & Seibert, J. (2012). Bias correction of regional climate model simulations for hydrological climate-change impact studies: Review and evaluation of different methods. *Journal of Hydrology*, 456, 12-29.
- Turco, M., Llasat, M. C., Herrera, S., & Gutiérrez, J. M. (2017). Bias correction and downscaling of future RCM precipitation projections using a MOS-Analog technique. *Journal of Geophysical Research: Atmospheres*, 122(5), 2631-2648.
- Turco, M., Sanna, A., Herrera, S., Llasat, M. C., & Gutiérrez, J. M. (2013). Large biases and inconsistent climate change signals in ENSEMBLES regional projections. *Climatic change*, 120(4), 859-869.

Responses to reviewer #3

Reviewer #3 (Highlight): *This study presented interesting results on the projection of fire activity into the future under various climate warming scenarios. I really like that the authors decided NOT to use standard IPCC RCP scenarios, but rather used scenarios of their own choosing, which puts the control in their hands.*

Response: We would like to thank the reviewer for his/her review. Each specific comment has been addressed in the manuscript as detailed below.

Comments

1. Referee's Comment: *Unfortunately, this is such a poorly written and organized paper that it really can't be published at this time. Here are some major problems I found in the paper:*

1. Poorly written. Many concepts, techniques, and statements are not clearly presented. Many terms are not cited with the appropriate literature.

Response: We have re-organized the manuscript to make it clearer. Moreover our manuscript was edited for English language usage, grammar, spelling and punctuation by one or more native English-speaking editors at Nature Research Editing Service. The editors focused on correcting improper language and rephrasing awkward sentences, using their scientific training to improved flow and sound, and a more professional and natural style of our manuscript.

2. Referee's Comment: *2. Poorly cited. There are literally thousands of papers on this topic for Australia and North America – where were those papers?*

Response: There are, indeed, many papers that analyse fire-climate relationships based on time series models or cross-sectional analysis, but none which consider adjustments in the fire-climate relationships in response to changed climate conditions. Nonetheless, we have included some of these papers as new references to better contextualize and discuss our work, reaching the limit of 70 references set by this journal.

3. Referee's Comment: *3. Wordy. The entire Results section could have been reduced by 40% by concentrating on the important messages.*

Response: We have accordingly reduced the Results and discussion section as presented in the previous version of the manuscript. However, we have introduced further analysis of the uncertainty associated with various sources: the RCM model uncertainty, the uncertainty associated with the model parameters when constructing the climate-fire models (Eqs. 1 and 2), the impact of the choice of predictor for the model of Eq. 2 to estimate the potential non-stationary response of BA to climate change, of the extrapolation of present-day relationships into the future (since future climate variables may exceed the historical extremes used to develop the empirical models), and of the use of bias-corrected or raw RCM data. We have thus extended the Results and discussion section considerably, adding new figures (2 to 7, some of which are not exactly new figures but modified, enriched ones) and a deeper discussion on the robustness of our results.

4. Referee's Comment: *4. Organize. Why are there methods in the Results section?*

One main concern in this manuscript is that it seems that the results don't tell us anything we don't already know, and the presentation of the results don't live up to the title and abstract.

Response: Following the journal's format guidelines, we present the Method section at the end of the manuscript, i.e. after the Results and discussion section, and therein provide not only outcomes, but also discussion. In order to make the presentation as clear as possible, the Results and discussion section also includes the main equations of the developed climate-BA models.

Addressing the Referee's main concern, the revised paper now includes a more in-depth description of the novelty of our study, which was probably poorly highlighted in the previous version. To the best of our knowledge, this is the first study to estimate the translation of Paris warming targets into impacts on future wildfires. This is also encouraged by the IPCC, which will devote an entire special issue to impacts at various global warming levels. This is also the first time that adjustment/adaptation analysis, which is widely used to assess the effects of climate change on different sectors (e.g. agriculture, see Butler and Huybers, Nat. Clim. Change 3, 68–72; 2013), has been applied to estimate BA potential changes, exploring the potentially non-stationary response of fire to climate under changed conditions.

Finally, we have now added a comprehensive analysis of the uncertainties of the predicted impacts, as explained above.

Specific comments

1. Referee's specific comment: *Title –How about getting rid of the “at 1.5, 2., and 3deg warming”?*

Response: We would prefer to maintain the original title in order to emphasize that our analysis is performed for different warming thresholds (on the basis of the Paris Agreement of December 2015 of the United Nations Framework Convention on Climate Change). However, since the journal limits the title to no more than 15 words, we changed it to “Climate change impacts on summer fires in Mediterranean Europe at 1.5, 2 and 3°C warming”.

2. Referee's specific comment: *Abstract – Fails to capture the essence of the study and its implications*

Response: We have modified the entire abstract in accordance with the comments of the reviewers.

3. Referee's specific comment: *First sentence is totally wrong – these are not the main factors, especially soil moisture – they are only some of the factors. Weather is identified as the most important factor.*

Response: We have re-organized the manuscript avoiding assertions that are not based on the existing literature (being appropriately referenced) or on our results.

4. Referee's specific comment: *Organization. I really didn't like the fact that Results were meshed with Discussion. The authors seemed to take liberties with space – much of the written text seems redundant or unneeded. In fact, the entire section could be reduced and reorganized to improve understanding. Some of the discussion material could be included in the introduction, for example.*

5. Referee's specific comment: *Terminology. Many terms throughout the document were not defined or cited.*

Response: Following the reviewers' comments, we have mostly re-organized the manuscript. Also, as indicated above, our manuscript was edited by one or more native English-speaking editors at Nature Research Editing Service. In particular, formatting experts of the Nature Research Editing Service have modified our page layout, text formatting, headings, title page, and figure placement to ensure agreement with the guidelines of Nature Communications.

6. Referee's specific comment: *Figures and Tables*

a. I think many of the figures could have been combine and some figures could be removed. The authors should concentrate on the figures that are meaningful to the results, not to understanding the methods.

b. Many of the tables need modifications as shown in the marked MS

Response: We have rethought the design of the figures and tables and the way we present the results, looking for a compromise between being detailed enough (to ensure its understanding by the wide and diverse readership of *Nature Communications*) yet as concise as possible. After deliberation, we feel that Figures 1 and 2 are still appropriate in the main manuscript to facilitate its understanding. Explanatory or illustrative figures, including flow charts or methodological schemes, are often included in similar papers (see e.g. Butler and Huybers, *Nat. Clim. Change* 3, 68–72; 2013; Canadell and Schulze, *Nat. Commun.* 5, 5282; 2014; Bowman, *et al. Nat. Ecol. Evol.* 1, 0058; 2017). Please also bear in mind that the journal encourages the inclusion of as much information as possible in the main manuscript and the avoidance of presenting important material as supplementary information.

Technical comments

1. Referee's technical comment: *Methods*

a. Why are there no methods in this paper? It looks like the methods are in the Results?

c. Having some methods at the end is very confusing.

Response: As already commented, we present the Method section at the end of the manuscript, i.e. after the Results and discussion section, following the journal's format guideline. Thus, in the Results and discussion section we provide the main model equations and basic methodological features to ensure the flow of the reading and to make the presentation of the results as clear as possible.

b. Equation 1 is badly described. Variables undefined, mixed scales.

Response: We have rewritten Eq.1 to make explicit the dependence on time and space of the variables. We have also done our best to explain it better in the text.

d. This section could be reduced.

Response: We have revised the Methods section putting effort into avoiding redundancies and striving to be brief. We also had the help of formatting experts of the Nature Research Editing Service to comply with the journal's requirements. Please note, however, that we have included further details of the steps undertaken with the new analyses performed in order to guarantee the reproducibility of our research.

Comments marked on the MS

Lines 22-23: "the sensitivity of such impacts to potential non-stationarity"

Referee's comment: *What the heck does this mean? Of course there is non-stationarity in climate-fire... why is this important?*

Response: In order to comply with the journal's requirements we have mostly rewrote the abstract and these lines have been eliminated in the revised paper.

Line 27: "modifications of climate-BA links under climate change"

Referee's comment: *What are the links? what variables?*

Response: In order to comply with the journal's requirements we have mostly rewrote the abstract and these lines have been eliminated in the revised paper.

Line 27: "found"

Referee's comment: *Past tense!*

Response: Done.

Line 29: "non-stationary"

Referee's comment: *this has not been defined*

Response: We have changed these lines to:

"Here, we estimate future summer burned area in Mediterranean Europe under 1.5, 2 and 3°C global warming scenarios and accounting for the possible modifications of the climate-fire relationships under changed climatic conditions".

Line 29: "Therefore, even under this assumption that led to lower impacts, only not surpassing the ambitious 1.5 C target consents to avoid to double BA in the future"

Referee's comment: *I have no idea what this sentence means? Clarify and rewrite*

Response: We have changed these lines to:

"Thus our results demonstrate that greater benefits would be achieved if warming is limited well below 2°C."

Line 43: "In order"

Referee's comment: *To delete it*

Response: Done.

Line 46: "impacts such as agriculture"

Referee's comment: *something wrong here -- agriculture is NOT an impact; the authors must have meant impacts TO..*

Response: We have revised the entire manuscript with the help of the Nature Research Editing Service. In particular, this phrase now reads:

"[...] and the relevant impacts on agriculture".

Lines 47-48: "However, the translation of ambitious warming targets into impacts on future wildfires remains to be studied"

Referee's comment: *in this region? because this has been done in many other areas*

Response: To the best of our knowledge, this is the first study to translate the 'Paris agreement' global warming targets into regional impacts on future wildfires, as recommended by the United Nations Framework Convention on Climate Change (UNFCCC), and the first to include a non-stationary approach to evaluate such impacts. Many papers have focused on climate change impacts on fires (we cite several, reaching the upper limit set by the journal for the list of references), but they consider changes for specific times periods, such as 2080-20100, under a particular representative concentration pathway (RCP) or emission scenario, and do not account for changes in the climate-fire relationships under future changed conditions.

Line 107: "SPEI transforms"

Referee's comment: *An index can't transform*

Response: We have reworded this:

"SPEI standardizes accumulated climatic balance".

Lines 109-110: "negative SPEI values identifying hot and dry situations"

Referee's comment: *Is this right? aren't high SPEI values hot dry?*

Response: It was right; negative SPEI values indicate a negative anomaly in the PRE-PET balance. To make this clearer, we changed these lines to:

"positive and negative values indicating wet and dry conditions, respectively".

Equation 1: "SPEI_{sc}"

Referee's comment: *is this the transformed SPEI?*

Response: The subscript sc made reference to the time scale at which SPEI is computed (we consider periods of 3, 6 and 12 months). In the revised version, we have tried to make it clearer and have modified the nomenclature of the equations.

Equation 1: "T"

Referee's comment: *the variable "T" is undefined*

Response: The term T in Eq. 1 is (and was) defined as the long-term temporal trend of the BA series resulting from both anthropic effects, such as the gradual increase in fire management effort, and environmental/climatic changes.

Line 126: "result"

Referee's comment: *To delete it*

Response: Done.

Lines 128-129: "[...]d to"

Referee's comment: *To delete them*

Response: Done.

Line 129: "estimate"

Referee's comment: *to change it to estimating*

Response: Done.

Lines 134-126: "T represents the linear temporal trends of the fire variable resulting from both anthropic effects (such as a gradual increase in fire management effort) and environmental/climatic changes."

Referee's comment: *I still do not know what T means?*

Response: We have made it clearer in the revised manuscript.

Lines 143-144: "Figure 1 show that the values of the parameter beta-2, representing the response of BA"

Referee's comment: *I stopped correcting grammatical mistakes from here to the end. I strongly suggest that an editor is used.*

Response: With the help of the Nature Research Editing Service, we have revised the manuscript thoroughly to correct any grammatical errors and improve its quality, clarity and flow.

Lines 147-148: "warmer and drier summers lead to larger fires"

Referee's comment: *this is NOT ground-breaking... thousands of papers have found this*

Response: We agree. Indeed, we consider this result 'straightforward', as is explicitly acknowledged in the paper. However, this relationship can be quite complex, and, as we show in Figure 2, the influence of drought conditions on BA is not constant across regions. In this sense, one major novelty of our study is the analysis of the non-stationarity of the climate-fire relationship under changed conditions, that is, how such a 'straightforward' relationship still holds or is transformed under future climate conditions.

Line 152: "In order"

Referee's comment: *To delete it*

Response: Done.

Line 152: "explain"

Referee's comment: *explained the most variation? How better?*

Response: This line has been deleted in the revised paper.

Line 153: "test"

Referee's comment: *past tense*

Response: Done.

Lines 167-172: "We interpret this spatial variation as a surrogate for potential non-stationarity in the BA-SPEI links. That is, these results suggests that the effects of climate change on BA is not totally obvious and it can be different from simple extrapolations using model-based short-term response of BA to climate, since changes in ecosystem structure and species composition can alter the climate-fire relationships."

Referee's comment: *this is nothing new.. thousands of papers have found this. several should have been cited here.*

Response: This adjustment/adaptation analysis, which is widely used to analyse the effects of climate change on agriculture (for example, Butler and Huybers, Nat. Clim. Change 3, 68–72; 2013), is here applied, for the first time, to estimate BA potential changes taking into account the change in the climate-fire relationship under future changed climatic conditions. To the best of our knowledge, there are many paper that analyse stationary climate-fire relationships based on time series models or cross-sectional analysis, but none which consider adjustments in the fire-climate relationship in response to climate change as done here. The list of references has been updated and extended (up to the limit of 70 items set by the journal guidelines) including a vast number of these previous works. Nonetheless, if we are mistaken and earlier studies on the non-stationary response of BA to climate under changed climate conditions do exist, we would greatly appreciate it if the reviewer would provide us with a list of references.

Lines 180-184: "Note that this similar adjustment/adaptation strategy is widely used to analyse the effects of climate change on agriculture (see, e.g.46,47), but, to the best of our knowledge, it has never been applied to study the impact on forest fires as done here."

Referee's comment: *I beg to differ. There are many papers on this*

Response: As commented above, there are certainly many analyses on climate-fire relationships based on time series models or cross-sectional analysis but, to the best of our knowledge, none which take the approach to develop the "non-stationary" model in the way this study has.

Figure 3

Referee's comment: *Is this a proper figure for Nature? shouldn't the subplots be labeled and referenced in the caption?*

Response: We have redone this figure, trying to improve its quality and including the appropriate labels in the subplots.

Figure 3, caption: "in"

Referee's comment: *to delete it*

Response: Done.

Figure 3, caption: "with respect to"

Referee's comment: *to delete it*

Response: Done.

Lines 180-184: "Similar results have been obtained."

Referee's comment: *to what? this is a repeat?*

Response: The sentence was certainly too brief to be understandable. It has been removed and the whole Results and discussion section has been reorganized following the reviewers' recommendations.

Lines 180-184: "Different PET estimation methods exist, ranging from the simplest approach (Thornthwaite) to more sophisticated approaches".

Lines 269-272: "It is worth noting that calculating the SPEI-BA correlation considering the Hargreaves or Penman estimation methods, led to very similar results (Supplementary Figures 4 and 5, respectively)."

Referee's comment: *Primarily because the Thornthwaite method is so bad and only good for coarse scale applications*

Response: Taking into account this comment, we have replaced the simple Thornthwaite approach to estimate potential evapotranspiration by a more appropriate method based on the Hargreaves estimation.

Reviewer #2 (Remarks to the Author):

The authors provided a detailed revision of their manuscript entitled "Impact of climate change on summer fires in Mediterranean Europe at 1.5, 2 and 3°C warming". Many comments were raised by the reviewers regarding the quality of the analysis and the writing. Overall, I appreciated the work done by the authors to answer the reviewers' comments and to provide a new version of the manuscript that was considerably revised. The scientific quality of their study has much improved compared to the previous version of the manuscript, in particular thanks to results provided by the uncertainty analyses, the better description of their modeling assumptions and equations and their inclusion of more theoretical background about the fire-climate relationships and references to previous work. Having said that, I have two main comments on this new version of the manuscript.

I think that the authors failed to properly address the issue related to the shift in the nature of the fire climate relationships that results from productivity gradients. I already mentioned this point in the previous round of reviews but maybe I was not clear enough. It is usually thought that the climatic controls on fire activity in the Mediterranean lie close to "tipping points", where relatively small changes in future climates could translate into drastic, and even divergent, shifts in fire activity because of productivity alterations. In other words, climate change might lead to a shift of the dominant constraint on fire activity, from a fuel moisture (in present times) to fuel amount (in future). This possibility is to quickly excluded by the authors (L52-54) whereas one main objective of this study is precisely to analyze the impact of vegetation on the fire-climate relationship.

Second, while I appreciate the efforts made by the authors to clarify their results, I think that the manuscript could be easily shortened (in particular the results and discussion section) by focusing on the main results and conclusions of their study. I am sure that such modifications would help to reach a broader readership and would strengthen the results of their study. Similarly, some figures could be moved in supplementary material.

Minor comments

*L23-24: Precise the nature of the modifications to the fire-climate relationship

*L28-29: the figures of projected BA for the other scenarios should be provided to justify this conclusion

*L52-56: This is a very strong hypothesis.

*L69-83: This paragraph could be easily shortened

* L87-92. This section is not very clear.

* L90-92: What about the more southern regions?

* L92-96: I am not sure this is necessary to mention the EFFIS dataset here. In addition, I wonder if this assertion is entirely justified since I though this dataset is based on MODIS remote sensing products, which are clearly known to have important biases, or Am I Wrong?

* L155-156: what "statistical analysis" provides a confirmation?

* L188-195: this could be moved in "Methods"

* L229-234 and L250-254: A possible shift in the nature of the fire-climate relationship should be discussed

L268-308: This all part could be easily shortened.

Reviewer #3 (Remarks to the Author):

Much better version of the manuscript. Much improved. A few comments:

1. The entire paper reads as if it was submitted to a statistics journal rather than Nature. It is dense with statistical jargon and terminology that may be lost on some of the readers. I strongly suggest that the authors emphasize the findings in a biological sense and use statistics to back the findings rather than present the statistics and leave it to the reader to interpret

2. the authors constantly start a topic sentence with "Figure X shows...". this is just lazy writing. The topic sentence should define the paragraph and the figure should be referenced parenthetically.

Responses to reviewer #2

Reviewer #2 (Highlight): *The authors provided a detailed revision of their manuscript entitled "Impact of climate change on summer fires in Mediterranean Europe at 1.5, 2 and 3°C warming". Many comments were raised by the reviewers regarding the quality of the analysis and the writing. Overall, I appreciated the work done by the authors to answer the reviewers' comments and to provide a new version of the manuscript that was considerably revised. The scientific quality of their study has much improved compared to the previous version of the manuscript, in particular thanks to results provided by the uncertainty analyses, the better description of their modeling assumptions and equations and their inclusion of more theoretical background about the fire--climate relationships and references to previous work. Having said that, I have two main comments on this new version of the manuscript.*

Response: We would like to thank the reviewer for this review. Each specific comment has been addressed in the manuscript as detailed below.

Comments

1. Referee's Comment: *I think that the authors failed to properly address the issue related to the shift in the nature of the fire climate relationships that results from productivity gradients. I already mentioned this point in the previous round of reviews but maybe I was not clear enough. It is usually thought that the climatic controls on fire activity in the Mediterranean lie close to "tipping points", where relatively small changes in future climates could translate into drastic, and even divergent, shifts in fire activity because of productivity alterations. In other words, climate change might lead to a shift of the dominant constraint on fire activity, from a fuel moisture (in present times) to fuel amount (in future). This possibility is to quickly excluded by the authors (L52--54) whereas one main objective of this study is precisely to analyze the impact of vegetation on the fire--climate relationship.*

Response: Following the editor and reviewer's recommendation, we reorganized the material in the Introduction section with a more in-depth description of the shift in fire--climate relationships.

2. Referee's Comment: *Second, while I appreciate the efforts made by the authors to clarify their results, I think that the manuscript could be easily shortened (in particular the results and discussion section) by focusing on the main results and conclusions of their study. I am sure that such modifications would help to reach a broader readership and would strengthen the results of their study. Similarly, some figures could be moved in supplementary material.*

Response: Following the editor and reviewers' recommendation, we reorganized the material in the Results section better highlighting the main results and conclusions of the study. However, we prefer to maintain all the figures in the revised manuscript bearing in mind that the journal encourages the inclusion of as much information as possible in the main manuscript and the avoidance of presenting important material in the form of supplementary information.

Minor comments

Referee's Comment: *L23--24: Precise the nature of the modifications to the fire--climate relationship*

Response: We changed these lines to:

“[...] accounting for possible modifications of climate-fire relationships under changed climatic conditions owing to productivity alterations”.

Referee's Comment: *L28-29: the figures of projected BA for the other scenarios should be provided to justify this conclusion*

Response: This conclusion is based on the previously reported burned area increases and thus our results fully support the statement of the Paris Agreement that limiting the temperature increase to 1.5 °C would "significantly reduce the risks and impacts of climate change". In addition, further expanding this conclusion will push the length of the abstract beyond the prescribed limits, so we prefer to avoid it.

Referee's Comment: *L52--56: This is a very strong hypothesis.*

Response: Actually, this is not our working hypothesis. We just introduce here the potential fire responses due to climate change, considering the nature of the fire climate relationships that results from productivity gradients. In addition, following the editor and reviewer's recommendation, we reorganized the material in the Introduction section with a more in-depth description of the shift in fire--climate relationships and we have changed these lines to:

“Only if the direct effect of climate change in regulating fuel moisture (e.g., drier and warmer conditions increase fuel flammability leading to larger fires) continues to be dominant with respect to the indirect effect on fuel load and structure (e.g., drier and warmer conditions limit fuel availability), fire risks will increase^{7,9,29-32} as the climate becomes warmer and drier^{33,34}.”

Referee's Comment: *L69--83: This paragraph could be easily shortened*

Response: Following the editor and reviewer's recommendation, we reorganized the material in the Introduction section and this paragraph has been shortened.

Referee's Comment: *L87--92. This section is not very clear*

Referee's Comment: *L90--92: What about the more southern regions?*

Referee's Comment: *L92--96: I am not sure this is necessary to mention the EFFIS dataset here. In addition, I wonder if this assertion is entirely justified since I though this dataset is based on MODIS remote sensing products, which are clearly known to have important biases, or Am I Wrong?*

Response: In order to comply with the editor and reviewer's recommendations, we rewrote this section and these lines have been eliminated in the revised version of the paper.

Referee's Comment: *L155--156: what "statistical analysis" provides a confirmation?*

Response: We have changed these lines to:

“The statistical analysis that follows provides a confirmation”.

Referee's Comment: *L188--L195: this could be moved in "Methods"*

Response: Following this comment, and in order to focus on the main findings in the section Results, these lines have been moved in the Methods section.

Referee's Comment: *L229--234 and L250--254: A possible shift in the nature of the fire--climate relationship should be discussed*

Response: We have added a discussion on this issue in the revised section on Results.

Referee's Comment: *L268--308: This all part could be easily shortened.*

Response: We shortened the manuscript wherever possible.

Responses to reviewer #3

Reviewer #3 (Highlight): *Much better version of the manuscript. Much improved. A few comments:*

Response: We have done our best to improve the manuscript following the reviewer's recommendations. Below we detail the modifications made to address the last comments.

Comments

1. Referee's Comment: *The entire paper reads as if it was submitted to a statistics journal rather than Nature. It is dense with statistical jargon and terminology that may be lost on some of the readers. I strongly suggest that the authors emphasize the findings in a biological sense and use statistics to back the findings rather than present the statistics and leave it to the reader to interpret.*

Response: Following the editor and reviewers' recommendation, we reorganized the material in the Results section to better highlight the main results and conclusions of our study.

2. Referee's Comment: *the authors constantly start a topic sentence with "Figure X shows...". this is just lazy writing. The topic sentence should define the paragraph and the figure should be referenced parenthetically.*

Response: We rewrote these sentences as recommended by the reviewer wherever appropriate.